# Biomarkers-based Biosensing and Bioimaging with Graphene for Cancer Diagnosis

**DOI:** 10.3390/nano9010130

**Published:** 2019-01-21

**Authors:** Hui Gu, Huiling Tang, Ping Xiong, Zhihua Zhou

**Affiliations:** Key Laboratory of Theoretical Organic Chemistry and Functional Molecule of Ministry of Education, Hunan Provincial Key Laboratory of Controllable Preparation and Functional Application of Fine Polymers, School of Chemistry and Chemical Engineering, Hunan University of Science and Technology, Xiangtan, Hunan 411201, China; 13762208533@163.com (H.T.); 13667410081@163.com (P.X.)

**Keywords:** biosensing, bioimaging, graphene, cancer diagnosis

## Abstract

At the onset of cancer, specific biomarkers get elevated or modified in body fluids or tissues. Early diagnosis of these biomarkers can greatly improve the survival rate or facilitate effective treatment with different modalities. Potential nanomaterial-based biosensing and bioimaging are the main techniques in nanodiagnostics because of their ultra-high selectivity and sensitivity. Emerging graphene, including two dimensional (2D) graphene films, three dimensional (3D) graphene architectures and graphene hybrids (GHs) nanostructures, are attracting increasing interests in the field of biosensing and bioimaging. Due to their remarkable optical, electronic, and thermal properties; chemical and mechanical stability; large surface area; and good biocompatibility, graphene-based nanomaterials are applicable alternatives as versatile platforms to detect biomarkers at the early stage of cancer. Moreover, currently, extensive applications of graphene-based biosensing and bioimaging has resulted in promising prospects in cancer diagnosis. We also hope this review will provide critical insights to inspire more exciting researches to address the current remaining problems in this field.

## 1. Introduction

Cancer, the second leading cause of death globally, accounts for 8.8 million deaths and possibly over 21 million by 2030 [1]. Because early discovery of cancer is difficult, almost all the patients are diagnosed with terminal cancer, which is awfully bad for cancer treatment. Consequently, the exploitation of novel and valid cancer diagnosis methods, with the capacity to elevate diagnosis accuracy of cancer at its early stage, is one of the key issues encountered by modern medicine.

In the past few years, nanodiagnostics including nanomaterial-based biosensing and bioimaging have attracted enormous interest throughout the scientific community because they brought about unparalleled advances in cancer discovery [2]. For a particular cancer, the most promising method is to detect its biomarkers or carcinoma cells. Normally, cancer biomarkers are cancer-related biological molecules in human tissues or fluids (urine, blood, saliva, and cerebrospinal fluids), such as nucleic acids, enzymes, proteins and small molecules [3]. However, in patients whose cancer is in the early stage, the expression level of the biomarker is usually at trace level [4]. Moreover, the biosystem is complex, containing numerous interference species. Therefore, early discovery of cancer proposes greater challenges on the sensitivity and selectivity for nanodiagnostics. Fortunately, recent advances in nanotechnology, enabling the abundant emergence of novel nanomaterials, have evoked the creation of advanced biosensing and bioimaging techniques for cancer diagnosis.

The discovery of graphene in 2004 has been accompanied by increasing research interests to explore this novel material for cancer diagnosis applications [5]. Graphene, a single layer of carbon atoms in a two-dimensional honeycomb lattice, possesses remarkable optical, electronic and thermal properties; mechanical and chemical stability; large surface area; and good biocompatibility, endowing its versatile application in nanoelectronics, quantum physics, and catalysis, reinforcing filler and energy research [6,7,8,9,10,11]. For nanomedicine, graphene has been developed as multiple effective devices with diverse functions of disease diagnostics and therapies. 

Actually, the detection of various cancer biomarkers with biosensing has already been summarized by several reviews [12,13,14,15,16]; specific recognition of biomarkers with bioimaging has also been published in some reviews [17,18,19,20]. However, these reviews are related to different diseases, and do not emphasize cancer diagnosis taking advantage of graphene.

In this review, we chose to survey graphene-based biosensing and bioimaging approaches concerning cancer biomarkers recognition in the past few years that could serve as alternative means for early cancer diagnosis. We chose this topic because the past researches clearly demonstrated that the discovery of graphene has put forward more possibilities in cancer diagnostics and therapies, which is now guiding modern medicine. In this review, firstly, different graphene nanomaterials and corresponding assemblies employed for biosensing and bioimaging are discussed. Then, the surface functionalization of recognition units such as antibody and aptamer are briefly summarized. Finally, the working principles as well as the recent advances of graphene-based biosensing and bioimaging for the specific diagnosis of cancer biomarkers are comprehensively reviewed. Overall, this review is devoted to presenting the recent advancements and future prospects of biomarker-based cancer diagnosis using graphene-based biosensing and bioimaging.

## 2. Graphene Nanomaterials 

The commonly used graphene includes 2D graphene films, 3D graphene architectures and GHs nanostructures [21].

### 2.1. 2D Graphene Films

2D graphene film is one of the most applicable forms of graphene. Polycyclic aromatic hydrocarbon defines a graphene molecule with a size of 1–5 nm, while nanographene is a graphene fragment with a size ranging from 1–100 nm [22,23]. In case of its size exceeding 100 nm, it is directly regarded as graphene. Thus, the commonly used 2D graphene films include graphene molecule, graphene nanoribbon, graphene quantum dot (GQD), and graphene sheet (GS) on the basis of their size scale (Figure 1). On the one hand, the oxidized form of graphene, graphene oxide (GO, Figure 2a), has rich oxygen-containing functional groups (e.g., −OH, C=O) on its surface, endowing it with an extremely large surface area. These functional groups facilitate the loading of functional species, which makes GO a consummate material for biosensing application [7,24,25]. On the other hand, the size of graphene does not influence these groups on its surface. Therefore, both nanographene and graphene keep most properties of GO. In addition to this, the nanographene has its own unique characteristics on account of its size effect such as fluorescence emission at ~500 nm through one or two photon excitation [26]. Herein, nanographene stands for graphene with sizes under 100 nm including graphene molecule, graphene nanoribbon and GQD, which are broadly used in bioimaging for their high quantum yields [26,27,28].

### 2.2. 3D Graphene Architectures

Recently, integrating 2D graphene films into macroscopic 3D porous networks has attracted great attention due to the 3D architectures with higher specific surface areas, stronger mechanical strengths, and faster mass and electron transport kinetics [30,31,32]. There are two common methods use for the fabrication of 3D graphene architectures: the self-assembly method and templated method. The self-assembly method tries to trigger graphene self-linking through a hydrothermal process or chemical reduction process [33,34,35,36]. The templated method tries to grow or fabricate graphene on a template such as Ni materials or polystyrene colloidal spheres, and eventually dissolves the template [37,38,39]. Obviously, 3D graphene architectures with abundant pores (Figure 2c,d,e) possess prominently increased active surface areas. The specific 3D graphene architectures, which favor the immobilization of functional units and accelerate the mass transfer rate of substrates [40], are usually applied in biosensing.

### 2.3. GHs Nanostructures

GHs nanostructures are constructed by graphene modified with other nanomaterials like carbon nanotubes (CNTs), polymers (Figure 3a), ionic liquids (ILs, Figure 3b), metals (Figure 3c), and metal oxides (Figure 3d). CNTs/graphene hybrids, with the advantageous properties of the two carbon allotropes, revealed faster electron transfer kinetics and increased specific surface areas [41]. Similarly, metal/metal oxide modified graphene, not only presented high catalytic properties of metal/metal oxide and a large surface area of graphene, but also overcame the shortcomings of metal/metal oxide such as low stability [6,42]. In addition, the embellishments of metal/metal oxide provide graphene with more possibilities to be fabricated with other functional species. Furthermore, polymers and ILs are usually assembled onto graphene to import new functional groups and enhance its biocompatibility.

## 3. Surface Functionalization with Recognition Units

Because both biosensing and bioimaging are involved with specific recognition events, it is of necessity here to declare the surface functionalization with active recognition units. The most widely used recognition units for biomarkers detection are aptamer, antibody and enzyme [47]. Typically, the recognition elements are normally linked onto the graphene surface through covalent and noncovalent interaction [48]. The covalent functionalization method is used to modify the target units onto graphene surface through a covalent bond. Noncovalent modification of recognition units onto graphene surface was involved with π–π interaction, Vander Wale force and hydrogen bonding. It is worth noting that the noncovalent functionalization method with no influence on graphene intrinsic structure, affords a weaker connection force compared with the covalent functionalization method [49].

### 3.1. Antibody

Antibody, an immunoglobulin, is a functional protein which can recognize a corresponding pathogen with high specific selectivity. Using this binding mechanism, antibodies can tag some detectable probes to discover cancer biomarkers. Antibodies modified onto graphene through the noncovalent absorption method has suffered from conformational changes as well as loss of function, which has been verified recently [50,51]. Therefore, covalent modification of an antibody onto graphene surface becomes the most effective and practical way. On one hand, oxygen-containing functional groups on GO facilitate the bonding of an antibody. On the other hand, functionalized graphene and GHs contain abundant useful groups, giving graphene remarkable ability to be covalently decorated with an antibody. For example, Hong et al. prepared branched polyethylene glycol (PEG)-modified nanographene and facilitated further covalent conjugation of various functional entities such as CD105. This kind of graphene can be specifically directed to the tumor neovasculature in vivo with the modification of CD105, a biomarker for vascular tumors [52].

### 3.2. Aptamer

Aptamers can distinguish among thousands of nucleotides and proteins in a short period of time. In addition to this, it can detect tiny differences of proteins with quite similar structures [53]. An aptamer with aromatic bases can be easily immobilized on the surface of graphene through noncovalent bonding without conformational changes. When encountering target biomarkers with more affinity, the aptamers tend to combine with the corresponding biomarkers and separate from from the surface of graphene. Moreover, the nucleotide base in an aptamer containing carbonyl and amino groups is available to be fabricated with various function groups like alkynyl, sulfydryl, carboxyl, and avidin. Their diversified groups can be covalently linked onto the surface of graphene or GHs. Azimzadeh et al. decorated graphene with Au moieties for functionalization of a captured SS-probe, which was used in direct detection of the breast cancer biomarker mRNA-155 [54]. On this platform, Oracet Blue can be accumulated onto the resulted hybrid of SS-probe-miRNA via intercalation, thus giving a reduction signal. Through this, targeted molecules can be specifically captured on the platform and quantized.

## 4. Graphene Based Cancer Nanociagnosis

The rapidly growing diagnosis methods for cancer are mainly divided into two streams: biosensing and bioimaging [55]. On these platforms, graphene plays its important roles. Firstly, graphene provides a natural biocompatible carrier to immobilize more specific recognition units, thus elevating the sensitivity of the diagnosis system. Secondly, graphene with excellent electric and optical properties offers an amplifying signal for sensing and imaging, which can further supply the lower detection limit to biomarkers detection. Thirdly, graphene with flexible functionalization characteristics can be integrated with a variety of desirable nanometerials to afford a multi-function system with a synergistic effect, achieving more precise and sophisticated diagnosis. In this article, we introduce the cancer diagnosis with graphene-based biosensing and bioimaging aiming at detecting cancer biomarkers. Protein, DNA, microRNA (mRNA), and certain small molecules can be used to define a kind of cancer and also the stage of the disease. For example, in the case of prostate cancer, prostate-specific antigen (PSA) is the biomarker usually used in the diagnosis of this pathology. For breast cancer, different mutational DNA such as BRCA1, BRCA2, HER2-neu, C-MYC, Cyclin D-1, and serum protein biomarker CA 153 are selected as its specific biomarkers. For liver cancer, the alpha fetoprotein (AFP) is the most used. Carcino embryonic antigen (CEA), a highly glycosylated protein in colorectal adenocarcinoma, is considered as one of the most important biomarkers. Furthermore, certain metabolized molecules like H_2_O_2_ and H^+^ level could also be regarded as important indicators in the progress of cancerization.

### 4.1. Biosensing

Biosensing for cancer diagnosis aims at designing effective biosensors to detect the specific signal variation when the target cancer biomarkers are combined with the recognition units. There mainly exist three methods to construct the biosensors: the direct method, competitive method and sandwich method. The direct method is to use a recognition unit-biomarker conjunction for the production of signal alteration without any label, which involves nonspecific absorption. The competitive method and sandwich method are involved in signal generation events by labels. These labeled signal elements include electric redox molecules like dyes; enzymes; luminol; metal nanostructures; and optical molecules like Alexa Fluor, Cy5, fluorescein isothiocyanate (FITC), fluorescein, tetramethylbenzidine (TMB), up-conversion phosphors (UCPs), and quantum dots (QDs). With the enhanced or weakened intensities of these labels generated by the combination between recognition units and targeted cancer biomarkers, the levels of these biomarkers can be revealed.

Graphene-based biosensing becomes an adaptable diagnostic method because of the practical role of graphene in electroanalysis and electrocatalysis, enabling cost effective, fast and in-situ analysis of biological samples with high accuracy and sensitivity [56]. Normally, biosensors are classified as electrochemical biosensors, optical biosensors, thermal-detection biosensors, ion-sensitive, field-effect transistor, and resonant biosensors according to its transducer unit [57,58], among which electrochemical biosensing and optical biosensing are most frequently used because of their simplicity. Herein, we introduce some prominent work using graphene-based biosensing to detect cancer biomarkers.

#### 4.1.1. Electrochemical Biosensing

Electrochemical biosensing based on the 2D graphene films, 3D graphene architectures and GHs nanostructures are widely used in various cancer biomarker detections due to their extremely large surface areas and abilities to interact with the various types of species referred to above. In the direct strategy, graphene are to provide a biocompatible surface to load more recognition units, thus contributing to sensitive biosensors. Verma et al. fabricated an electrochemical immunosensor with Au NPs-rGO as the transducer matrix for noninvasive detection of oral cancer biomarker interleukin-8 (Figure 4) [59]. Chen et al. designed an electrochemical DNA biosensor based on Ag NPs-3D graphene foam for sensitively and specifically detecting the biomarker CYFRA21-1 DNA from a non-small cell in lung cancer [60]. Electrochemical Biosensing is an effective and fast strategy for early clinical diagnosis of cancer, which has been further confirmed by other similar works [61,62,63,64,65,66].

However, constructing biosensors through sandwich strategy attracts more attention because of its higher specificity towards biomarkers detection. Normally, there are graphene-based biosensors and graphene-based probe-amplifying biosensors. The graphene-based biosensor is to employ graphene as an electronic matrix to immobilize recognition units. The first example was developed by Lin group (Figure 5). Chitosan–graphene serves as a substrate material to load Ab1, which can be specifically combined with targeted AFP. Another horseradish peroxidase (HRP) labeled Ab2 conjunct with AFP was then attached to produce a detectable signal [67]. In this work, greatly enhanced sensitivity for the AFP was obtained because of the dual signal amplification strategy of “sandwich” immunoreactions. After that, many similar and effective studies were reported [68,69,70,71,72,73,74,75], among which it is worth noting that a microfluidic paper-based electrochemical biosensor for the multiplexed detection of multi-biomarkers was achieved [73]. This is another big advance, because it means that spontaneous diagnosis of more than one kind of specific cancer could be possibly acquired. The graphene-based probe-amplifying biosensor is to employ labeled graphene as an amplifying unit to generate a higher detectable signal [70,76,77,78,79,80]. For example, Wang et al. developed a novel electrochemical multiplexed immunoassay for simultaneous determination of AFP and CEA. In this strategy, the label probes were obtained by connecting two primary antibodies (anti-AFP and anti-CEA) onto a graphene surface [79]. Furthermore, recently there have sprung up electrochemical biosensors using graphene as both a substrate matrix and probe amplifier to doubly enhance detection sensitivity [81,82,83,84,85].

In Table 1, we summarized the recent studies for the cancer biomarker detection with graphene-based electrochemical biosensors. As demonstrated in Table 1, the detection limits differ from fM to nM, which further reveals the potential ability of graphene towards cancer biomarkers monitoring.

#### 4.1.2. Optical Biosensing

Optical biosensing has become a prevailing technology in biomedical diagnostics, which contains fluorescence, colorimetric, interferometric, and surface plasmon resonance (SPR) biosensors. The optical tunability of nanographene and its derivatives are advantageous in optical biosensing. Meanwhile, recognition molecule are involved in a reversible link–delink process on the graphene surface, which is the basis of designing specific optical biosensors. Qu et al. developed a naked-eye readout colorimetric immunoassay for PSA detection based on the intrinsic peroxidase activity of GO [115]. Compared with absorbance-based biosensing, fluorescence biosensing is much more sensitive (~1000 times higher) to enabling single-molecule detection [116]. As shown in Figure 6, NGO was applied to absorb dye-labeled peptide nucleic acid by Ryoo et al., which exhibited quenched fluorescence signal [117]. When target mRNA was added, the nucleic acid demonstrated higher affinity towards mRNA than NGO, which led to the recovery of the signal. The detection limit was as low as ~1 pM, and three different mRNAs in a living cell were simultaneously monitored. The dynamic changes of mRNA expression level are associated with disease type and stage, which allowed the constructed mRNA bioimaging to diagnose the early stage of cancer cells with high precision and discrimination power. Similar works to discover biomarkers of prostate and bladder cancer were reported by Kanara group [118] and Xia’s group [119].

As for the protein biomarker, many prominent studies by optical biosensors have been reported. One example is demonstrated in Figure 7. The fluorescence of the probes (DNA-Ag NCs and DNA-Ag/Au NCs) were quenched by introducing GO, while more competitive interactions of cancer biomarkers mucin 1, CEA and CA 125 facilitated the recovery of the fluorescence [120]. Meanwhile, protein biomarkers were detected by Chiu et al. through carboxyl functionalized GO (GO–COOH)-based SPR. The high affinity of GO–COOH and anti-CK19 protein resulted in higher sensitivity for the detection of CK19 protein in non-small cell lung carcinoma compared to a conventional Au-based SPR chip [94]. In Table 2, we summarized the recent studies for the cancer biomarker detection with graphene-based optical biosensors.

### 4.2. Bioimaging

Bioimaging enables the morphological features of organs to be correlated with pathological symptoms, which is important for finding biomarkers and subsequently using them for the diagnosis of diseases [138]. GQD and nanographene have been increasingly applied in bioimaging due to their unique optical properties such as near-infrared photoluminescence, magnetic resonance, characteristic Raman bands, and tunable fluorescence. Furthermore, graphene-based bioimaging provides a much better choice for detecting cancer in vivo regardless of the above biosensors used only in vitro. A number of groups have devoted efforts to explore bioimaging techniques for cancer diagnosis in vitro and in vivo with graphene and its derivatives.

#### 4.2.1. In Vitro Imaging

In-vitro cancer-related bioimaging includes direct cancer cells imaging and targeted cancer cells imaging. Numerous studies have demonstrated that GQD and nanographene with good optical properties are available to get access into cancer cells. Fan demonstrated that boron-doping GQDs gave rise to rich fluorescence owing to their peculiar interaction with the surrounding media. This kind of boron-doping GQD can be further used to detect biomarkers through cellular imaging [139]. Liu et al. reported that propidium iodide (PI)-GO can enter into the nuclei and stain the DNA/RNA of living human breast cancer cells MCF-7 [140]. In this study, PI alone without GO cannot cross the cellular membrane, revealing the fact that GO is an efficient PI delivery nanomaterial for live cell staining. Dong et al. firstly synthesized multi-layer GQD to label the cell nucleus [141]. Then, Hela cells [142,143,144,145], A-549 cells [146], breast cancer cell [147,148], gastric carcinoma cells [149] and hepatocellular cells [150,151], and U251 Glioma cells [152] were also successfully labeled by nanographene. All these studies suggested that nanographene with low cytotoxicity and excellent biocompatibility is an eco-friendly material in bioimaging. However, these graphene-based materials are non-selective and can enter both cancerous and healthy cells, which is hard to further apply in in-vivo cancer diagnosis.

Then there emerged targeted cancer cells imaging. The targeted cancer cells imaging could be achieved by modifying graphene with recognition units with promoted cancer-specific detection. The earliest studies [41,153] by Dai’s group demonstrated their synthesized PEGylated NGO (Figure 8a) is a potential material to be biologically applied in imaging in vitro and in vivo. PEGylated NGO was conjugated with B-cell-specific antibody Rituxan (anti-CD20) to afford NGO-PEG (NGO-PEG-anti-CD20) which was employed for selectively recognition of B-cell lymphoma cells (Figure 8b). As shown in Figure 8c,d, the NGO-PEG-anti-CD20 was selectively adsorbed on positive Raji B-cell surfaces and not on negative CEM T-cells. After that, Dai’s group attached another targeting peptide bearing the Arg-Gly-Asp motif to NGO-PEG, which can be used as an effective probe to be swallowed by U87MG cancer cells [154]. More recently, Huang’s group fabricated Apt MUC1-conjugated Au NPs/GO (Apt MUC1-Au NPs/GO, Figure 9a–d). The bioimaging results in Figure 9 e–g show that the Apt MUC1-Au NPs/GO had good ability to effectively and selectively find MUC1 units on tumor cell membranes [155,156]. Other recognition units such as folic acid (FA) [143,157,158,159,160], Hyaluronic acid (HA) [161,162,163,164], protein [165,166,167], peptides [168,169] and aptamer [170,171,172] have been used to targeted cancer cells imaging in recent years. For example, FA molecules were attached to the RGO-Ag NPs by physisorption for specifically targeting cancer cells with folate receptors (FRs) [159]. The Raman signals of the FA in live cancer cells were detected by confocal Raman spectroscopy at 514 nm excitation. RGO-Ag NPs-FA material generated a strong signal to indicate the overexpression of FA levels in Hela cells. These targeted cancer cells imaging are of high significance and can be further applied for in-vivo cancer diagnosis.

#### 4.2.2. In-Vivo Imaging

Since targeted cancer cells imaging was achieved, in-vivo imaging has become an alternative method to diagnose cancer. Especially the feasibility of carbon dots as a fluorescence contrast agent for in-vivo optical imaging was successfully reported in 2009 by Yang et al [173]. Just 1 year after that, the first success of in-vivo cancer imaging and photothermal therapy through graphene intravenous administration declared the great promise of graphene in cancer diagnosis and treatment [174]. There ultimately brought the prosperity of graphene-based application in in-vivo imaging with high selectivity. Among these significant studies, GQD or NGO were additionally conjugated with HA [175,176], aptamer [177], peptide [178], antibody [52,179], and FA [168,180] to ensure selective targeting of cancer sites in vivo. For example, Vallis’s group explored anti-HER2 antibody (trastuzumab)-conjugated NGO for imaging (Fig. 10) [179]. The anti-HER2 antibody was first radiolabeled with 111In-benzyl-diethylene-triaminepentaacetic acid (BnDTPA) via pp-stacking (Figure 10a), which was then i.p. injected into mice. In two HER2-overexpressing murine models of human breast cancer, high tumor-to-muscle ratio was achieved (Figure 10b,c), which demonstrated the effectiveness of this selective probe. Additionally, Hwang’s group successfully explored a novel theranostic platform based on GO-PEG-FA for fluorescence imaging of cancer [181]. After 24 h post intravenous injection, the near-IR luminescence from the internalized GO-PEG-FA was clearly visualized with a higher tumor-to-background ratio (TBR). This study demonstrated that GO-PEG-FA was a very good theranostic nanomaterial with great potential to be used in clinical. In addition, it can act as a fluorescent cellular marker as well as a dual modal reagent for the effective destruction of solid tumors.

## 5. Conclusion and Perspectives

The past few years have witnessed significant developments of graphene-based nanomaterials in cancer diagnosis applications. In this article, we summarized the great potential of graphene-based materials for cancer diagnosis. Three kinds of graphene-based materials have been introduced: 2D graphene films, 3D graphene architectures and GHs. The fabrication of a variety of recognition units including aptamers, antibodies and enzymes enables graphene to identify specific cancers. In addition, graphene in turn makes cancer identification more sensitive, selective and accurate on account of its elegant versatility and outstanding properties. Two main methods of cancer monitoring including biosensing and bioimaging are extensively introduced both in vitro and in vivo. As demonstrated in this article, the research on cancer diagnosis applications of graphene has seen dramatic progress and is expanding rapidly, yet it is still in its infancy.

The booming advances made in this area are currently exciting and encouraging; the challenges, however, are also huge and still need to be overcome. Much more attention should be given to the issues of its clinical practicability. The major challenge facing graphene-based cancer diagnosis lies in the huge variations presented in experimental designs among researches. Dawidczyk et al. pointed out in 2014 that the lack of universal rules in preclinical trials have obstructed a systematic comparison of these studies [181]. Therefore, setting up a uniform standard is indispensable for the development of graphene-based devices with high practicality to discovery of cancer. 

For in-vitro cancer diagnosis, diagnosis of cancer at its earliest stage often requires the detection of biomarkers with very low concentrations in body fluids, which is a complex system including different kinds of interference species. Therefore, biomarkers with low specificities result in a high risk of false–positive signals. Moreover, a single biomarker cannot serve as a perfect cancer screening tool to achieve the purposes of diagnosis, prognosis and therapy. It is urgent to develop effective graphene-based devices with multifunction, which can detect multi-biomarkers simultaneously to promote the accuracy of cancer discovery. Furthermore, design of simpler, faster, smaller graphene-based devices for the monitoring of cancer biomarkers is a thriving research and development area which elevates the potential applications graphene in cancer diagnosis. For in-vivo cancer diagnosis, better understanding of the behaviors of graphene in vivo is remarkably important to investigate their toxicology. Although NGOs, especially PEG-NGO, have been reported to be biocompatible in biomedical applications, its potential long-term toxicity and nonbiodegradable nature are still a major concern for clinical use. More studies on the potential acute and chronic effects of graphene and its metabolism/excretion in the body, are urgently required. In addition, the application in drug delivery and cancer therapy is one of the hottest fields of graphene-based research. Exploitation in multipurpose probes and the integration of both diagnosis and treatment of graphene are extremely promising for elevating the efficiency of diagnosis and therapy.

Overall, as has been discussed, graphene with outstanding physical and chemical features defines its prospective progress in the area of cancer diagnosis, appearing to put a solid milestone on the discovery of early cancer. Researchers are continuing to conquer the difficulties above and will eventually develop graphene-based devices capable of clinical application.

## Figures and Tables

**Figure 1 nanomaterials-09-00130-f001:**
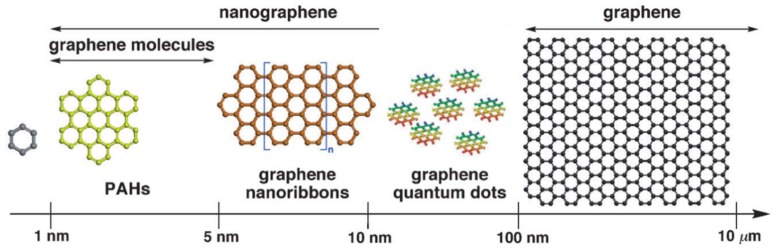
Schematic illustration of 2D graphene film defined according to their size scale. Reproduced from [22] (PAHs, graphene and graphene nanoribbons), with permission from Wiley, 2012. Reproduced from [29] (GQD), with permission from The Royal Society of Chemistry, 2012.

**Figure 2 nanomaterials-09-00130-f002:**
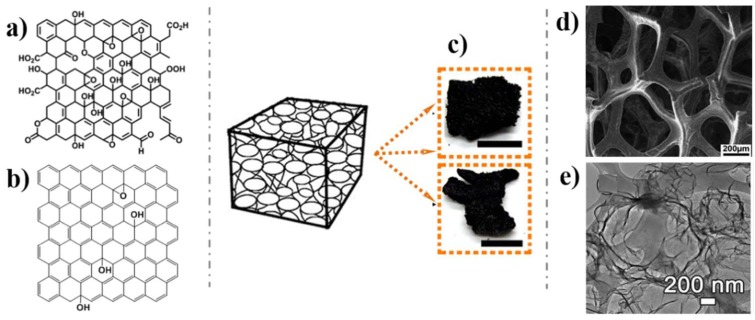
Schematic illustration of 2D graphene film defined according to their oxidation functional groups: (**a**) GO; (**b**) RGO; 3D graphene architectures: (**c**) Photographs; (**d**) Scanning electron microscope (SEM) image; (**e**) Transmission electron microscope (TEM) images. Reproduced from [40] (**a**), with permission from The Royal Society of Chemistry, 2014; Reproduced from [37] (**c**,**e**), with permission from American Chemical Society, 2017; Reproduced from [38] (**d**), with permission from American Chemical Society, 2012.

**Figure 3 nanomaterials-09-00130-f003:**
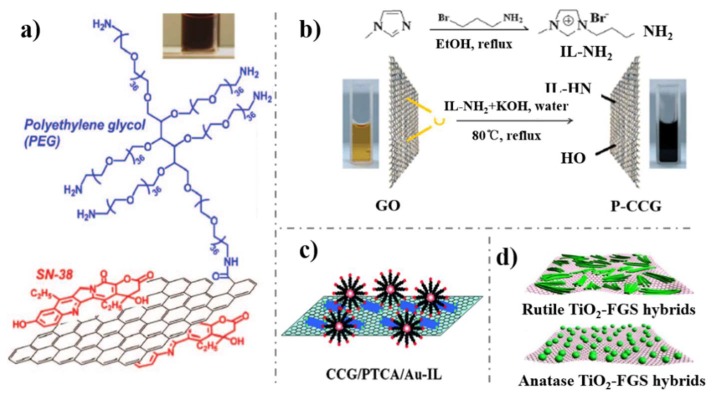
**(a**) Schematic drawing of PEG-NGO; (**b**) IL-RGO; (**c**) Graphene-PCTA-Au-IL; (**d**) TiO_2_-graphene hydrids nanostructures. Reproduced from [43] (**a**), with permission from American Chemical Society, 2008; Reproduced from [44] (**b**), with permission from The Royal Society of Chemistry, 2009; Reproduced from [45] (**c**), with permission from The Royal Society of Chemistry, 2009; Reproduced from [46] (**d**), with permission from American Chemical Society, 2009.

**Figure 4 nanomaterials-09-00130-f004:**
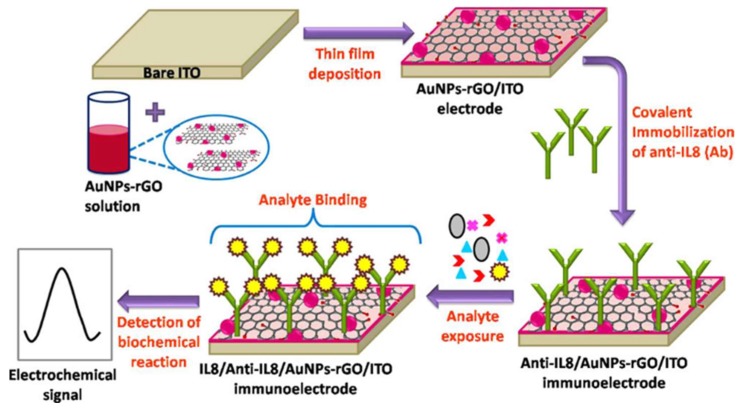
Schematic of fabrication of Au NPs-rGO-based immunoelectrode for immunosensing application. Reproduced from [59], with permission from American Chemical Society, 2017.

**Figure 5 nanomaterials-09-00130-f005:**
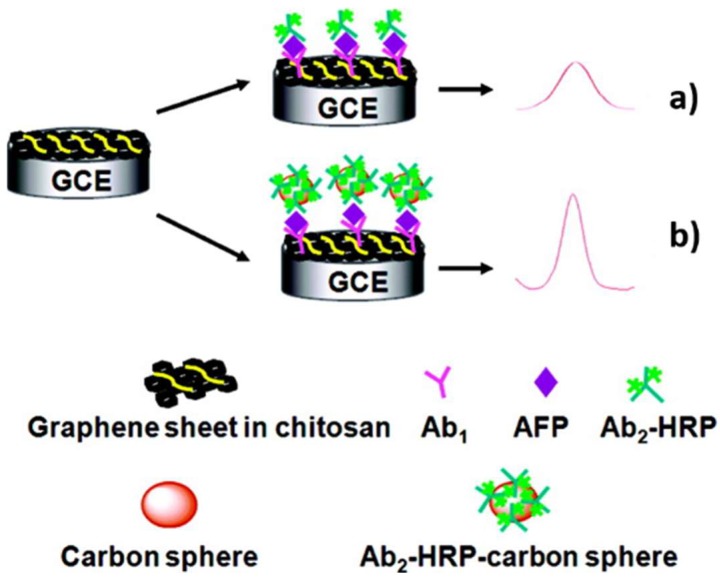
Schematic illustration of the detection principles of HRP-Ab2/AFP/Ab1/GS-CHI/SPCE (**a**) and HRP-Ab2-CNSs/AFP/Ab1/GS-CHI/SPCE; (**b**) using a signal amplification strategy at the graphene sheets biosensor platform. Reproduced from [67], with permission from American Chemical Society, 2010.

**Figure 6 nanomaterials-09-00130-f006:**
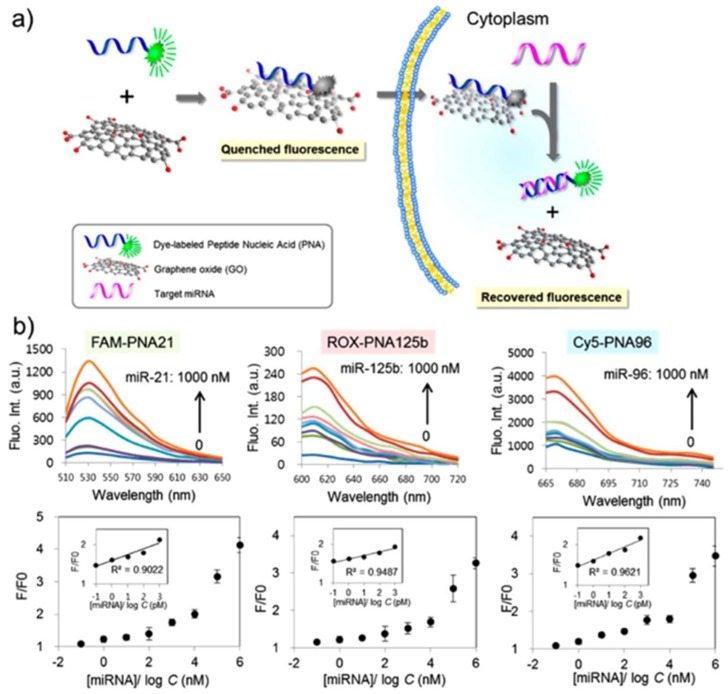
Scheme of strategy for an mRNA sensor based on NGO and PNA, and emission spectra and fluorescence images obtained for multiplexed mRNA sensing in vitro. (**a**) The fluorescence signal gets recovered when the fluorescent dye-labeled probes initially adsorbed onto the surface of NGO detach from NGO and hybridize with a target mRNA. (**b**) Three PNA probes: PNA21, PNA125b and PNA96 were designed for three different mRNA detections and prepared as conjugated with three different fluorescent dyes: FAM, ROX and Cy5. Reproduced from [117], with permission from American Chemical Society, 2013.

**Figure 7 nanomaterials-09-00130-f007:**
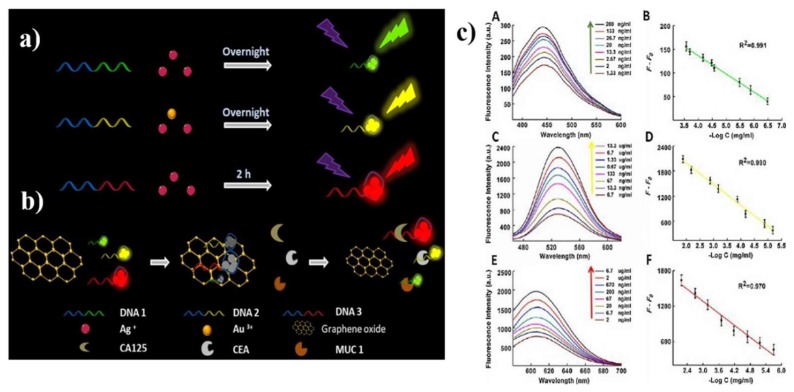
(**a**) Schematic illustration of synthesizing DNA-Ag NCs and DNA-Ag/Au NCs; (**b**) schematically demonstrating the mechanism for assaying MUC1, CEA and CA125; (**c**) fluorescence spectra of the assaying model in the presence of varying concentrations of MUC1 (**A**), CEA (**C**) and CA125 (**E**), the linear relationships between F-F0 and the concentrations of MUC1 (**B**), CEA (**D**) and CA125 (**F**), respectively. Reproduced from [120], with permission from Elsevier, 2018.

**Figure 8 nanomaterials-09-00130-f008:**
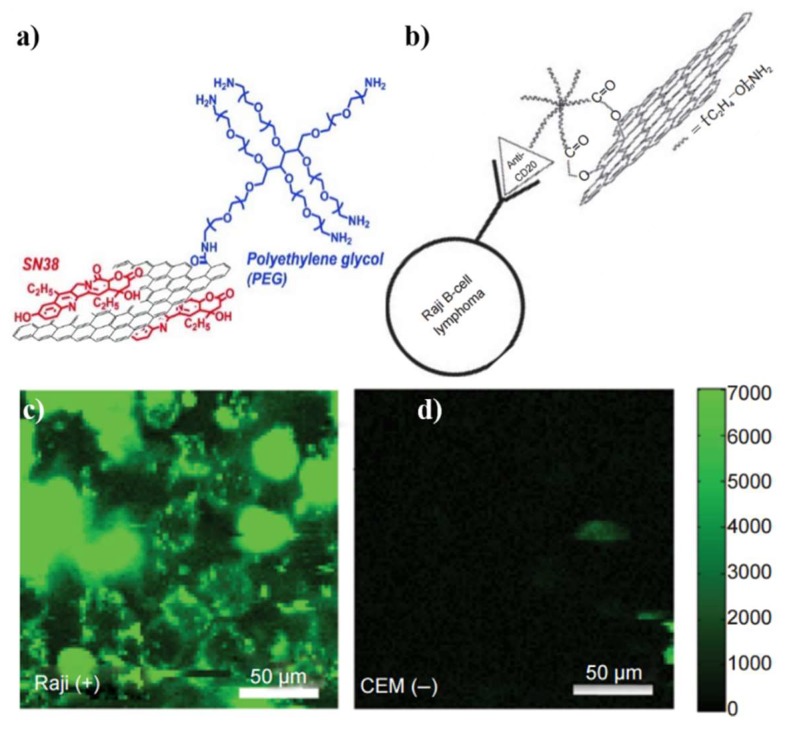
Schematic illustration of (**a**) PEG-NGO; (**b**) the selective binding and cellular imaging of PEG-NGO conjugated with anti-CD20 antibody, Rituxan; NIR fluorescence image of CD20 positive Raji B-cells (**c**) and CD20 negative CEM T-Cells (**d**) treated with the NGO PEG Rituxan conjugate. The scale bar shows the intensity of total NIR emissions (in the range 1100–2200 nm). Images are false-colored green. Reproduced from [43] (**a**), with permission from American Chemical Society, 2008. Reproduced from [153] (**b**–**d**), with permission from Springer Nature, 2008.

**Figure 9 nanomaterials-09-00130-f009:**
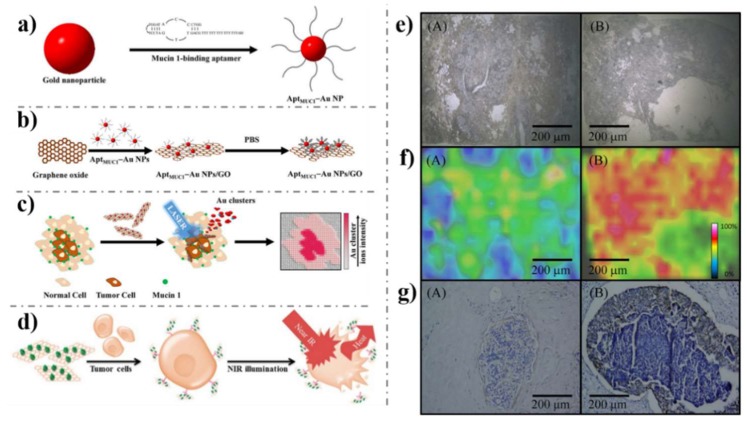
Schematic representation of (**a**) the preparation of MUC1-binding aptamer-modified gold nanoparticles (Apt MUC1–Au NPs) and (**b**) their conjugation to graphene oxide (Apt MUC1–Au NPs/GO) for (**c**) tumor tissue imaging by monitoring Au cluster ions when coupled with laser desorption/ionization mass spectrometry; (**d**) its application with NIR laser irradiation for photothermal therapy of cancer cells; (**e**) optical images; (**f**) laser desorption ionization mass spectrometry (LDI-MS) images of the [Au1]^+^ intensity distributions in tissues after incubation with Apt MUC1–Au NPs/GO for 1 h. (**g**) Immunostaining of MUC1 by MUC1 antibody (VU4H5) in tissues. (**A**) Normal breast and (**B**) tumor breast tissues. Reproduced from [155] (**a**–**c**,**e**–**f**), with permission from Nature, 2015; Reproduced from [156] (**d**), with permission from American Chemical Society, 2015.

**Figure 10 nanomaterials-09-00130-f010:**
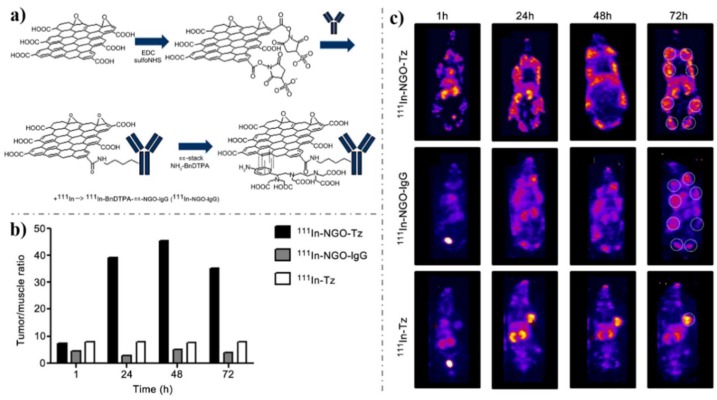
(**a**) Synthesis of ^111^In-NGO-IgG or ^111^In-NGO-Tz. (**b**) Quantification of the biodistribution of the mice in **c**. (**c**) Representative whole-body SPECT images (MIPs) of the biodistribution of ^111^In-NGO-Tz, ^111^In-NGO-IgG, or ^111^In-Tz in spontaneous tumor-bearing BALB/neuT mice at 1, 24, 48, or 72 h i.p. injection. Tumors are located in mammary fat pads, with metastases in the axial and inguinal lymph nodes (white dashed circle in 72 h images). Tumor burden is likely different in each animal due to the nature of the spontaneous tumor model. Reproduced from [179], with permission from Elsevier, 2013.

**Table 1 nanomaterials-09-00130-t001:** Summary of innovative electrochemical biosensors for cancer biomarkers detection.

Material	Target biomarker	Detection limit	Ref.
CdS nanocrystals/graphene oxide-AuNPs	p53	4 fg/ml	[86]
AgNPs in PANI and N-doped graphene	miRNA-21	0.2 fM	[87]
3D porous network graphene aerogel	CA 153	0.03 mU/mL	[88]
ss-DNA amino-functionalized GQD	mRNA-25	95.0 pM	[89]
AuPtPd/rGO	H_2_O_2_	2 nM	[90]
rGO/ordered mesoporous carbon/Ni-oxytetracycline metallopolymer NPs	EGFR exon21 L858R point mutation	120 nM	[91]
AFP aptamer GO	AFP	3 pg/mL	[92]
CysA/Au NSs/GQDs	CA 153	0.11 U/mL	[93]
GO-COOH	cytokeratin 19	1 fg/mL	[94]
Cubic CeO2/RGO	Cyfra-21-1	0.625 pg/mL	[95]
Anti-CEA/PBSE/Graphene/Cu	CEA	0.23 ng/mL	[96]
Ab1/P5FIn/erGO, Ab2/GQDs@AuNP	CEA	3.78 fg/mL	[97]
GS-PS67-b-PAA27-Au	PSA	40 fg/mL	[98]
Hemin-GS/PdNPs	PSA	8 pg/mL	[99]
Pd@Au@Pt/COOH-rGO	PSACEA	8 pg/mL 2 pg/mL	[100]
MWCNTs-COOH/rGO	CA 125	0.5 nU/mL	[101]
ss-DNA/3D GF/Ag NPs	CYFRA21-1	10 fM	[60]
anti-CYFRA21-1/3D graphene@Au NPs	CYFRA 21-1	100 pg/mL	[102]
CuS/rGO	CA 153	0.3 U/mL	[103]
ERBB2c, CD24c modified Au NPs/GO	HER2	0.16 nM, 0.23 nM	[104]
Au/ZnO/RGO	AFP	0.01 pg/mL	[105]
anti-PSA/Au NPs/GO	PSA	0.24 fg/mL	[106]
Au–S-GS	CEA NMP22	25 fg/mL 30 fg/mL	[68]
Fc-GNs/aptamer/BSA/DNA/Au-CdS flower-like 3D assemblies	PSA	0.38 pg/mL	[107]
rGO/Fe_3_O_4_@GO	PSA PS membrane antigen	15 fg/mL 4.8 fg/mL	[108]
graphene coated SPR chip	FAP	5 fM	[109]
sulfur-doped rGO	8-hydroxy-2’-deoxyguanosine	1 nM	[110]
FAD/Th/rGO-PAMAM/Aunano	CYC VEGF_165_	63.9 pM 38.4 pM	[111]
TB-Au-Fe3O4-rGO	AFP	2.7 fg/mL	[112]
rGO–metal nanocomposites	ErbB2	<1 fM	[113]
Anti-CEA/PDA-rGO	CEA	0.23 pg/mL	[114]

**Table 2 nanomaterials-09-00130-t002:** Summary of innovative optical biosensors for cancer biomarkers detection.

Probe	Target biomarker	Detection limit	Ref.
Aptamer-GO probe	C-myc/TK1/actin	0.26 nM	[121]
Ag NPs/GO	PSA	0.23 pg/mL	[122]
DNA aptamer GO	Exosome	21000 particles/μl	[123]
PT-DNA/GO	folate receptor	0.44 pM	[124]
boron-doped GQD	alkaline phosphatase	10±5 cells/mL	[125]
ssDNA/GQD	CYFRA21-1	0.3 μU/mL	[126]
Hemin-GS	Telomerase	60 cells/mL	[127]
DNA-GO nanocomposites	Flap structure-specific endonuclease 1	0.38 pM	[128]
DNA/GO	epidermal growth factor receptor	390 pg	[129]
DNA/GO	hCG	20 mIU/mL	[130]
GQD@MnO_2_	glutathione	83 nM	[131]
telomerase/miR-21 oligonucleotides/GO	Telomerase/mRNA 21	10 pM/10 HeLa cells	[132]
GO/MnO_2_/fluorescein	glutathione	1.53 μM	[133]
DNA/GO	CEA	28.5 fg/mL	[134]
DNA/GO	telomerase	2.7 cells	[135]
DNA/GO	telomerase	30 cells	[136]
PT-Man@GO, PT-Gal@GO	lectin	7.9 nM	[137]

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
