# Peer review of "Biomarkers-based Biosensing and Bioimaging with Graphene for Cancer Diagnosis"

_nanomaterials, 2019, doi:10.3390/nano9010130_

Round 1

Reviewer 1 Report

General comments

The submitted paper consists in a review about the employement of graphene-based nanomaterials in the biosensing and bioimaging applications for cancer diagnosis.

The topic of the present work is worthy of investigation and well matches the aim and scope of Nanomaterials. It is strongly suggested to improve and properly expand the Conclusions section with final considerations and future developments.

Moreover, a deep English grammar and language revision is strongly recommended.

More details and specific remarks and suggestions are reported below point by point.

Introduction

- The Introduction section is well organised.

- Concerning the graphene applications, also its use as reinforcing filler has to be reported, citing, for example “Neat and GNPs loaded natural rubber fibers by electrospinning: manufacturing and characterization, Materials & Design 88 (2015): 1109–1118”.

2. Graphene Nanomaterials

Please replace Fig. 1 with Fig. 2.

2.2. Graphene hybrids nanostructures

Please replace Fig.3 with Fi. 4.

3. Surface Functionalization with Recognition Units

- All this paragraph needs to be corroborated with suitable literature references.

3.1. Antibody

-          The Authors should specify all the acronyms the first time they used them. For example, PEG has to be extensively reported in this paragraph.

3. Graphene Based Cancer Nanociagnosis

Please replace ‘3.’ with ‘4.’

4.2.2. In Vivo imaging

Please replace Fig. 17 with Fig. 11.

3. Conclusion

This section has to be properly expanded. The Authors should add further final considerations and future developments.

Author Response

Point 1: Introduction: Concerning the graphene applications, also its use as reinforcing filler has to be reported, citing, for example “Neat and GNPs loaded natural rubber fibers by electrospinning: manufacturing and characterization, Materials & Design 88 (2015): 1109–1118”.

Response 1: Per the reviewer’s suggestion, the advance of application of graphene as reinforcing filler has been documented in the revised manuscript. And the related paper has been properly cited.

Point 2: Graphene Nanomaterials: Please replace Fig. 1 with Fig. 2.

2.2. Graphene hybrids nanostructures: Please replace Fig.3 with Fig. 4.

Response 2: Per the reviewer’s suggestion, Fig. 1 and 3 has been replaced with Fig. 2 and 4 in the revised context.

Point 3: Surface Functionalization with Recognition Units: All this paragraph needs to be corroborated with suitable literature references.

Response 3: Per the reviewer’s suggestion, literature references as ref 47-49 have been properly cited, thus firmly corroborating the statements of this paragraph.

Point 4: 3.1. Antibody: The Authors should specify all the acronyms the first time they used them. For example, PEG has to be extensively reported in this paragraph.

Response 4: Per the reviewer’s suggestion, all the acronyms used in the manuscript were re-checked thoroughly and specified at the first time. Accordingly, “polyethylene glycol” was specified for “PEG” at the first time.

Point 5: 3. Graphene Based Cancer Nanodiagnosis: Please replace ‘3.’ with ‘4.’

Response 5: Per the reviewer’s suggestion, 3 has been replaced with 4 in the revised paper.

Point 6: 4.2.2. In Vivo imaging: Please replace Fig. 17 with Fig. 11.

Response 6: Per the reviewer’s suggestion, Fig. 17 has been replaced with Fig. 11 in the revised paper.

Point 7: 3. Conclusion: This section has to be properly expanded. The Authors should add further final considerations and future developments.

Response 7: Per the reviewer’s suggestion, conclusion has been rewritten and properly expanded.

First, we summarized the main idea of this review article. (The past few years have witnessed significant development of graphene-based nanomaterials in cancer diagnosis applications. In this article, we summarized the greata potential of graphene-based materials for cancer diagnosis. Three kinds of graphene-based materials have been introduced: 2D graphene films, 3D graphene architectures or GHs. Fabrication of a variety of recognition units including aptamer, antibody and enzyme enables graphene identify specific cancers. And graphene in turn makes cancer identification more sensive, selective and accurate on account of its elegant versatility and outstanding properties.   Two main methods of cancer monitoring including biosensing and bioimaging are extensively introduced both in vitro and in vivo. As demonstrated in this article, the research on cancer diagnosis applications of graphene has seen dramatic progress, and is expanding rapidly, yet still in its infancy.)

Second, the main challenges of this area has been presented. (The booming advances made in this area are currently exciting and encouraging, the challenges, however, are also very huge and remain to be overcome. Much more attention should be drawn to the issues of its clinical practicability. The major challenge facing graphene-based cancer diagnosis lies in the huge variations presented in experimental designs among researches, Dawidczyk et al. has pointed in 2014 that the lack of universal rules in preclinical trials obstructed a systematic comparison of these studies prevents and thus limited further advancements in the field. At this point, more efforts should be spared on the uniformity in methods to exploit graphene-based devices aiming at various biomarkers detection.

Third, the future developments of this area has been proposed. (For in vitro cancer diagnosis, diagnosis of cancer at its earliest stage often requires detection of the biomarkers with very low concentrations in body fluids which is complex systems including different kinds of interference species. Therefore, the biomarkers with low specificities raises a high risk of false-positive signals. Moreover, single biomarker cannot serve as perfect cancer screening tool to achieve the purposes of diagnosis, prognosis and therapy. Develop novel versatile graphene devices with multifunctionalities and multimodalities, which provides the possibility of simultaneous measurements of various biomarkers with higher accuracy. Design of simpler, fast, smaller graphene-based devices for monitoring of cancer biomarkers is a thriving research and development area, which elevates the potential applications graphene in cancer diagnosis. For in vivo cancer diagnosis, better understanding of behaviors of graphene in vivo is remarkably important to investigate their toxicology. Although NGO especially PEG-NGO has been reported to be biocompatible in biomedical applications, its potential long-term toxicity and nonbiodegradable nature are still the major concern for clinical use. More studies on the potential acute and chronic effects of graphene and its motablism/excretion in the body, are urgently required. And the application in drug delivery and cancer therapy is one of the hottest fields of graphene-based research. Exploitation in multipurpose probes integration of both diagnosis and treatment of graphene, are extremely promising for elevate the efficiency of diagnosis and therapy.) 

Reviewer 2 Report

Dear Authors

The article is well written and give deep understanding of graphene, structure and composites, however there are already articles published on this topic. Also, there is a few data for imagining and diagnostic portion of cancer. It can be published after major revisions. Below there are few comments to improve this article. 

1- Section 3.1 line 2 the word is pathogen with or pathogen? please correct it.

2- Please include one, two specific examples of aptamer and enzyme-based recognition of cancer biomarker etc. with grapheme in section 3.2 and 3.3.

3- It will be more effective if you can put one graph, flowchart or table to show the graphene based electrochemical sensor, optical and others sensors, that can present the cancer diagnostic/imaging  work (authors gave only one example of each but will be better if it cover maximum data in table form or chart etc.).   

4- There are already review on this topic (you can see below, red lines), what is the difference between your review and others?

5- Please check the English mistakes (miswriting etc.) also the abbreviations. 

Minors:

The author’s name are wrong.

1-Graphene-Encapsulated Nanoparticle-Based Biosensor for the Selective Detection of Cancer Biomarkers

2- Graphene: The Missing Piece for Cancer Diagnosis?

Author Response

Point 1: Section 3.1 line 2 the word is pathogen with or pathogen? Please correct it.

Response 1: Per the reviewer’s suggestion, the mistake has been corrected in the revised paper. Actually, it is pathogen.

Point 2: 2-    Please include one, two specific examples of aptamer and enzyme-based recognition of cancer biomarker etc. with graphene in section 3.2 and 3.3.

Response 2: Per the reviewer’s suggestion, a specific example of SS-probe-based recognition of cancer biomarker mRNA-155 with graphene as matrix has been exampled in the revised paper. And Section 3.3 has been deleted. The level of many small molecules vibrate substantially in the progress of tumor and these molecules are sometimes considered as cancer biomarkers, using enzymes to detect their noteworthy variation in level is clever. However, enzyme serves as a recognition unit to diagnosis of cancer has been rarely studied, because the levels of small molecules are easy to be affected by various physiological process and other pathological events. It is hard to identify cancer through detection of single small molecule through enzyme.

3-    It will be more effective if you can put one graph, flowchart or table to show the graphene based electrochemical sensor, optical and others sensors, that can present the cancer diagnostic/imaging work (authors gave only one example of each but will be better if it cover maximum data in table form or chart etc.).  

Response 3: Per the reviewer’s suggestion, some tables are added to present the advances of graphene-based cancer diagnosis in the past two years. And also, figure 1 is a brief chart to show how biosensing and bioimaging serve as effective methods to detect cancer biomarkers.

4-    There are already review on this topic (you can see below, red lines), what is the difference between your review and others?

Response 4: The paper 1 (Graphene-Encapsulated Nanoparticle-Based Biosensor for the Selective Detection of Cancer Biomarkers) actually is a research article. The authors demonstrated a novel strategy for the fabrication and application of rGO encapsulated NP-based biosensor for selective and sensitive detection of key biomarker proteins (HER2 and EGFR) for breast cancer. The selective detection of HER2 and EGFR was carried out by functionalizing the rGO-NPs with monoclonal antibodies against HER2 or EGFR. This a good example of biosensing aiming at cancer biomarker detection.

The paper 2 (Graphene: The Missing Piece for Cancer Diagnosis?) is really a good review article for readers to understanding the graphene based biosensor to detect characteristic cancer biomolecules. What they focused on are introduction of the recent advances in graphene-based biosensors with better performance for earlier cancer detection.

In this review, we chose to survey graphene-based biosensing and bioimaging approaches concerning cancer biomarkers recognition in the past few years that could serve as alternative means for early cancer diagnosis. We focused on introduction of two main methods by taking advantages of the good properties inherited in graphene to discover cancer, aiming at in vitro diagnosis and in vivo diagnosis respectively. Moreover, the latest advances in the past two years has been concluded in the revised article. Therefore, in term of this aspect, we have reasons to believe that the topic we reviewed here would be interesting for the readers who are interested in the further sduty of the graphene-based biosensing and bioimaging for developing techniques of cancer diagnosis.

5- Please check the English mistakes (miswriting etc.) also the abbreviations.

Response 4: Per the reviewer’s suggestion, the English mistakes and the abbreviations have been thoroughly checked and corrected.

6- The author’s name are wrong.

Response 6: Per the reviewer’s suggestion, the mistake has been corrected in the revised paper. Ping has been substituted as Ping Xiong.

Reviewer 3 Report

This review manuscript needs extensive revision due to the poor preparation of each figure and the requirement of extensive English correction. 

Author Response

Point 1: This review manuscript needs extensive revision due to the poor preparation of each figure and the requirement of extensive English correction.

Response 1: Per the reviewer’s suggestion, we added two tables to more clearly present the latest advances in the past two years in the revised article. The figures shown in the paper are typical examples to explain how the kind of method used for biomarker detection. And the English mistakes and the abbreviations have been thoroughly checked and corrected.

Round 2

Reviewer 1 Report

The Authors have followed all Reviewers’remarks. Now the paper looks very improved.

Author Response

Thanks a lot for your help!

Reviewer 2 Report

Dear authors

the manuscript is improved.

Thanks

Author Response

Thanks a lot for your help!

Reviewer 3 Report

Although the authors changed the manuscript, it still needs extensive revision. Most of the figures cannot readable because they have very small letters and there are still English issues across the text.  

Author Response

Response to Reviewer 3 Comments

Point 1: Although the authors changed the manuscript, it still needs extensive revision. Most of the figures cannot readable because they have very small letters and there are still English issues across the text. 

Response 1: Per the reviewer’s suggestion, we updated all the figures in the revised manuscript. And after thoroughly reading the manuscript, we briefly correct the English mistakes. All the revised portions have been marked in red in the revised manuscript.

This manuscript is a resubmission of an earlier submission. The following is a list of the peer review reports and author responses from that submission.